# Biophysical and X-ray structural studies of the (GGGTT)$_3$GGG G-quadruplex in complex with *N*-methyl mesoporphyrin IX

**Linda Yingqi Lin**[1], **Sawyer McCarthy**[1], **Barrett M. Powell**[1], **Yanti Manurung**[1], **Irene M. Xiang**[1], **William L. Dean**[2], **Brad Chaires**[2], **Liliya A. Yatsunyk**[1]*

**1** Department of Chemistry and Biochemistry, Swarthmore College, Swarthmore, Pennsylvania, United States of America, **2** Structural Biology Program JG Brown Cancer Center, University of Louisville, Louisville, Kentucky, United States of America

* lyatsun1@swarthmore.edu

**Data Availability Statement:** Data have been deposited in the RCSB Protein Data Bank at https://www.rcsb.org/structure/6P45 (DOI 10.2210/

## Abstract

The G-quadruplex (GQ) is a well-studied non-canonical DNA structure formed by G-rich sequences found at telomeres and gene promoters. Biological studies suggest that GQs may play roles in regulating gene expression, DNA replication, and DNA repair. Small molecule ligands were shown to alter GQ structure and stability and thereby serve as novel therapies, particularly against cancer. In this work, we investigate the interaction of a G-rich sequence, 5'-GGGTTGGGTTGGGTTGGG-3' (T1), with a water-soluble porphyrin, *N*-methyl mesoporphyrin IX (NMM) via biophysical and X-ray crystallographic studies. UV-vis and fluorescence titrations, as well as a Job plot, revealed a 1:1 binding stoichiometry with an impressively tight binding constant of 30–50 µM$^{-1}$ and $\Delta G_{298}$ of -10.3 kcal/mol. Eight extended variants of T1 (named T2 –T9) were fully characterized and T7 was identified as a suitable candidate for crystallographic studies. We solved the crystal structures of the T1- and T7-NMM complexes at 2.39 and 2.34 Å resolution, respectively. Both complexes form a 5'-5' dimer of parallel GQs capped by NMM at the 3' G-quartet, supporting the 1:1 binding stoichiometry. Our work provides invaluable details about GQ-ligand binding interactions and informs the design of novel anticancer drugs that selectively recognize specific GQs and modulate their stability for therapeutic purposes.

## Introduction

DNA typically exists *in vivo* in a well-defined, right-handed double helix. However, it can also adopt several other secondary structures, particularly when in its single-stranded form, such as at the end of telomeres and during replication, transcription, and DNA repair [1]. G-quadruplex (GQ) DNA is a well-studied non-canonical DNA structure formed by the π-π stacking of G-quartets. Each G-quartet is formed by a planar arrangement of four guanines connected by cyclic Hoogsteen hydrogen bonding. GQs are further stabilized by cations, notably K$^+$, that bind in the center between each pair of G-quartets [2]. GQs readily form *in vitro*, with their thermodynamic stability in physiological buffers often rivaling that of double-stranded DNA.

[pdb6P45/pdb](https://www.rcsb.org/structure/6PNK)) and https://www.rcsb.org/structure/6PNK (DOI 10.2210/pdb6pnk/pdb).

**Funding:** This work is supported by the National Institutes of Health [grant number 1R15CA208676-01A1 to LY; GM077422 to BC and WD] and Henry Dreyfus Teacher-Scholar award to LY. This work is based upon research conducted at the Northeastern Collaborative Access Team beamlines, which are funded by the National Institute of General Medical Sciences from the National Institutes of Health (P30 GM124165). The Eiger 16M detector on 24-ID-E beam line is funded by a NIH-ORIP HEI grant (S10OD021527). This research used resources of the Advanced Photon Source, a U.S. Department of Energy (DOE) Office of Science User Facility operated for the DOE Office of Science by Argonne National Laboratory under Contract No. DE-AC02-06CH11357. The funders had no role in study design, data collection and analysis, decision to publish, or preparation of the manuscript.

**Competing interests:** The authors have declared that no competing interests exist.

The next-generation sequencing based G4-seq strategy was used to identify >700,000 sequences with quadruplex-forming potential (QFP) in the human genome [3], notably in telomeres, oncogene promoters, and untranslated regions of mRNA. G4-seq was recently extended to other organisms [4]. Certain sequences with high QFP are highly conserved between species and localize to functional genomic regions [5], suggesting an evolutionary pressure to conserve these structures. GQs may serve a variety of essential biological functions. They are strongly associated with cancer due to their inhibitory roles during replication, transcription, and DNA repair, causing DNA damage and genomic instability [6]. Bioinformatics studies suggest that 37–94% of human genes contain sequences with high QFP near their promoter regions [7], and that oncogenes and regulatory genes are more G-rich than housekeeping genes. Any cellular process that involves G-rich DNA in a single-stranded form could potentially be regulated by GQs. Transient formation of GQs was demonstrated even when duplex DNA structure has not been disrupted [8]. Mounting experimental evidence firmly establishes GQ DNA as a viable therapeutic target for cancer andother human diseases [6].

The DNA sequence studied in this work, 5'-(GGGTT)₃GGG-3' (T1), folds into a stable parallel GQ structure. It has been previously studied by us (under the name THM) [9] and Largy *et al.* (under the name 222) [10]. No structural information on T1 or T1 in complex with small-molecule ligands is available, however. Here, we investigate the interaction of T1 with *N*-methyl mesoporphyrin IX (NMM, Fig 1), a water-soluble porphyrin with a distinctive central methyl group [11]. We have demonstrated previously that NMM is highly selective for parallel GQ topology over other DNA structures and can serve as a fluorescent probe for GQ DNA [9, 12]. Our studies also revealed that NMM greatly stabilizes T1 by 23 ± 2 ˚C at 2 eq. [9]. However, the mode and strength of T1-NMM binding has not yet been characterized.

Here we set out to investigate T1 binding to NMM via biophysical methods and X-ray structural studies. To increase our chances for successful crystallization, we designed a variety of constructs based on the genomic context of T1. We used BLAST [13] to align T1 with a number of genomes. The T1 sequence occurs 11 times in the human genome in seven distinct regions across six chromosomes including within the zinc finger protein 292 gene; on the complementary strand of the plexin A4 gene; within C-type lectin domain family 2 member A gene; carboxypeptidase M gene, *fas* binding factor 1 gene; and LOC107984696 non-coding RNA (S1 Table). T1 may also be involved in regulating *β*-secretase 2 (BACE2), a gene implicated in Alzheimer's disease [14]. In addition, the T1 sequence is found at position 168,273 in the *Tetrahymena thermophila* genome in the telomeric region. However, it does not represent the common telomeric motif, (GGGGTT)ₙ (crystallographic studies on this consensus motif are underway in our laboratory). Recently, a protein that binds parallel telomeric GQs with $K_d$ of 11.5 μM was discovered in *T. thermophila* [15], thereby providing compelling evidence for the biological relevance of such GQ structures. Finally, the T1 sequence is found in the bacteria *Neisseria meningitidis*, *Neisseria gonorrhoeae*, the bird pathogen *Escherichia coli* strain APEC O78, and repetitively in *Paenibacillus* [16, 17].

For crystallization studies, we designed native constructs, T2 –T6, to contain the full T1 sequence expanded by 1–3 nt in the 5', 3', or both directions. We designed the T7 construct to prevent the dimerization observed in T1 by adding 5' and 3' thymine overhangs. Finally, we designed two constructs (T8 and T9) to promote efficient crystal packing via the formation of intermolecular Watson-Crick base pairs by adding cytosine and guanine at the 5' and 3' ends. We choose the C-G base pair for its strong hydrogen bonding. All construct sequences are listed in Table 1.

The importance and urgency of our work is underlined by the relative scarcity of GQ-ligand structures. The first crystal structure of a GQ-ligand complex was determined only in 2003 [18].

**Fig 1. Structure of *N*-methyl mesoporphyrin IX (NMM).**

**Table 1. DNA sequences studied in this work and their thermodynamic parameters in 5K buffer.**

| Name | Sequence 5′ → 3′ | ε, mM$^{-1}$cm$^{-1}$ | $T_m$, ˚C | $\Delta T_m$, ˚C* | $\Delta H$, kcal/mol | Hysteresis, ˚C | Oligomerization |
|------|------------------|------------------------|-----------|-------------------|----------------------|----------------|-----------------|
| T1 | GGGTTGGGTTGGGTTGGG | 173.0 | 57.7 ± 0.3 | - | 77 ± 2 | 3.3 | D |
| T2 | GGGTTGGGTTGGGTTGGG**GT** | 191.6 | 61.3 ± 0.4 | 3.6 | 83 ± 3 | 3.2 | D, smeary |
| T3 | **G**GGGTTGGGTTGGGTTGGG**GT** | 201.7 | 58.4 ± 0.3 | 0.7 | 51.9 ± 0.9 | 2.5 | M+D |
| T4 | **GG**GGGTTGGGTTGGGTTGGG**GT** | 211.8 | 60.1 ± 0.3 | 2.3 | 50.6 ± 0.8 | 2.2 | M+D |
| T5 | **TGG**GGGTTGGGTTGGGTTGGG**GT** | 219.3 | 58.8 ± 0.3 | 1.1 | 50.8 ± 0.8 | 2.8 | M |
| T6 | **TGG**GGGTTGGGTTGGGTTGGG**GTT** | 227.4 | 57.9 ± 0.4 | 0.2 | 53.0 ± 0.4 | 2.9 | M |
| T7 | **T**GGGTTGGGTTGGGTTGGG**T** | 189.0 | 52.0 ± 0.3 | -5.8 | 48.6 ± 0.1 | 2.2 | M |
| T8 | **C**GGGTTGGGTTGGGTTGGG**G** | 189.6 | 56.4 ± 0.3 | -1.3 | 46 ± 3 | 2.9 | M |
| T9 | **G**GGGTTGGGTTGGGTTGGG**C** | 189.2 | 52.9 ± 0.3 | -5.0 | 38.1 ± 0.5 | 3.1 | M+D |

All sequences are based on T1, with additional nucleotides indicated in bold. Melting temperature, enthalpy of unfolding, and hysteresis were determined via CD melting experiments, while oligomerization states were determined via PAGE and AUC (M–monomer, D–dimer).

*relative to T1

Since then, only 38 unique X-ray and 29 NMR GQ-ligand structures have been reported in the Protein Data Bank (PDB). These numbers were obtained by searching PDB for "G-quadruplex Ligand", refining the search parameters by "DNA" and either "X-ray Diffraction" or "Solution NMR", and manually examining all entries keeping those that are unique. The limited number of the reported structures hinders drug discovery [19], which requires detailed knowledge of molecular architectures and drug binding sites. Driven by this gap in knowledge, we determined crystal structures of T1-NMM and T7-NMM to 2.39 and 2.34 Å, respectively, as well as characterized stoichiometry, strength, and thermodynamic parameters of the T1-NMM interaction. This work furthers our understanding of ligand structural features that are essential for selective GQ binding, providing both molecular-level insights and atomic coordinates to inform the design of GQ-targeting anticancer drugs.

## Materials and methods

### DNA, ligand, and buffers

Lyophilized oligonucleotides were purchased from Integrated DNA Technologies (IDT; Coralville, IA) with standard desalting purification. DNA was hydrated in doubly distilled water to 1–2 mM and stored at -80˚C. Extinction coefficients for all sequences were obtained using IDT's OligoAnalyzer 3.1 and DNA concentration was determined from UV-vis spectra collected at 95 ˚C. The full list of DNA sequences used in this work, along with their extinction coefficients and thermodynamic parameters, can be found in Table 1. To induce GQ formation, DNA was diluted into the desired buffer, heated at 90–95˚C for 5–10 minutes, cooled slowly to room temperature over 4 hours, and equilibrated at 4˚C overnight. NMM stock was prepared in doubly-distilled water and its concentration was determined using an extinction coefficient of $1.45 \times 10^5$ M⁻¹cm⁻¹ at 379 nm [20]. All biophysical experiments were performed in **5K** buffer consisting of 10 mM lithium cacodylate pH 7.2, 5 mM KCl, and 95 mM LiCl. Crystallization trials were performed in **20K** buffer consisting of 10 mM lithium cacodylate pH 7.2 and 20 mM KCl.

### UV-vis spectroscopy

All UV-vis experiments were performed on a Varian Cary 300 UV-vis spectrophotometer equipped with a Cary temperature controller (± 0.3 ˚C error). Spectra were collected from 220–349 nm for DNA and 352–480 nm for NMM at 0.5 nm intervals with a 0.1 s averaging time, 300 nm/min scan rate, 2 nm spectral bandwidth, and automatic baseline correction.

**UV-vis titration.** NMM samples were prepared in 1 cm methyl methacrylate cuvettes to target an absorbance of ~0.5 (1000 μL of 3–4 μM NMM). DNA stock solutions were prepared at 90–170 μM to achieve a final [DNA]/[ligand] ratio of at least 1.5. To maintain constant NMM concentration throughout the titration, stock DNA samples contained an equivalent amount of NMM, which was added after annealing. During the experiment, DNA was titrated into NMM in increasing increments. The resulting samples were equilibrated for 2 min before UV-vis spectra were collected. The titration continued until no further changes were observed in at least three consecutive UV-vis spectra. The volume of DNA added (30–100 μL total), $\lambda_{max}$, and absorbance at $\lambda_{max}$ was monitored throughout the titration. Data was processed using singular value decomposition followed by direct fit as described below. Reported $K_a$ values represent the average of at least four consistent trials.

**Singular Value Decomposition (SVD).** The SVD method, described by Qu and Chaires [21], is a matrix-based method that allows for global processing of titration data using all wavelengths, thereby providing a strong advantage when compared to traditional fitting of data from only one (or several) wavelengths. First, we imported the data into MATLAB as a matrix

*M* consisting of the signal at each wavelength (columns) for every addition (rows). We decompose this matrix into three matrices *U*, *S*, and *V* such that *M* = *USV*$^T$ (where T stands for transpose). Matrix *U* consists of columns which represent the signal from each individual component; the diagonal matrix *S* contains singular values that serve as weighing factors; and matrix *V*$^T$ contains amplitude column vectors that indicate how much of each component is present at every addition of DNA. DNA concentration was plotted against a vector in *V* to generate a binding curve. While each column in *V* represents the same binding event, we used the second column to fit UV-vis titration data and the first column to fit fluorescence titration data because they generated curves that looked most representative of a binding event.

**Analysis of binding isotherms.** Binding isotherms were fit to a simple *DNA + ligand → DNA-ligand complex* binding model with 1:1 binding stoichiometry (see eqs. 4 and 5 in reference [21]). This method can also be used for higher stoichiometries by assuming equivalent and independent binding sites (although this treatment is likely an oversimplification). In such cases, the concentration of DNA binding sites was set equal to the concentration of DNA multiplied by the appropriate stoichiometric ratio. Binding models with stoichiometries of 1:1, 2:1, and 1:2 were tested in this work. The concentration of binding sites at each addition was used as the independent variable and the appropriate column vector from matrix *V* was used as the dependent variable. NMM concentration, as experimentally determined via UV-vis, was either kept constant or allowed to float if it provided a higher quality fit. A refined NMM concentration was accepted only if it deviated less than 20% from the measured value. Fitting the data yielded the most probable binding stoichiometry and the binding constant, $K_a$. The chosen models had the lowest stoichiometry, low $K_a$ error (less than 20%), random residuals, and a fit curve closely matching the data upon visual inspection. All data fitting was performed in GraphPad Prism 4.

**Thermal Difference Spectra (TDS).** TDS were obtained by subtracting UV-vis scans taken at 4 ˚C after 5 minutes of equilibration from scans taken at 95 ˚C after 10 minutes of equilibration. In principle, the low and high temperature limits are defined by the temperatures at which the DNA is (mostly) folded and unfolded, respectively. GQs have a characteristic trough in their TDS spectra around ~296 nm [22].

**Job plot.** A Job plot is a continuous variation analysis method that allows for model-independent determination of binding stoichiometry [23]. The method requires two UV-vis titrations involving equal concentrations of DNA and ligand (~2.50 μM). In the first titration, cuvettes containing 1 mL of NMM were placed into the sample and reference cells. DNA was titrated into the sample cell while an identical volume of buffer was titrated into the reference cell in increasing increments from 20 to 200 μL. In the second titration, a cuvette containing 1 mL of DNA was placed in the sample cell while a cuvette containing 1 mL of buffer was placed in the reference cell, and equivalent volumes of NMM were titrated into both cuvettes in increasing increments from 50 to 200 μL. Absorbance difference measured at the wavelengths of minimum and maximum absorbance were plotted against the mole fraction of NMM. The peak or trough in the Job plot indicates the mole fraction of NMM bound to DNA and thereby reports on the stoichiometry of the DNA-ligand complex. The presented data reflect results from three consistent trials.

## Fluorescence (FL) titration

Fluorescence experiments were conducted on a Photon Technology International QuantaMaster 40 fluorometer at 20.0 ˚C. Data were collected in the emission range of 560–720 nm with 2 nm slit widths, 0.5 nm step size, and 0.5 s integration time. The isosbestic point determined through UV-vis titrations (391 nm for the T1-NMM complex) was used as the excitation

wavelength. NMM in a methyl methacrylate fluorescence cuvette (~1.0 μM, 1500 μL) was titrated with small increments of 130–190 μM DNA to target a final [DNA]/[NMM] ratio of at least 1.5. The total volume of DNA added (40–60 μL), $\lambda_{max}$, and intensity at $\lambda_{max}$ was monitored throughout the titration. Data were analyzed in the same way as UV-vis titration data and reported $K_a$ values represent the average of five trials.

## Circular Dichroism (CD) scans and melts

All CD experiments were conducted on an Aviv 435 circular dichroism spectrophotometer equipped with a Peltier thermal controller (± 0.3˚C error) in 1 cm quartz cuvettes. DNA samples were annealed at ~5 μM alone or with 2–3 eq. of NMM in 5K buffer. CD scans were taken at 20 ˚C from 220–330 nm with a 1 s averaging time, 2 nm bandwidth, and 1 nm step. Three scans were collected first for the buffer and then for each sample in the corresponding cuvettes. CD data were processed as described in our earlier work [24].

CD melting experiments were conducted from 25–95 ˚C with a 1 ˚C step, 1 ˚C/min temperature rate, 15 s averaging time, and 5 s equilibration time. CD signal at 262 nm, the wavelength corresponding to maximum signal in CD scans, was monitored as a function of temperature. Melting temperatures, $T_m$, were determined via two methods. The first method involves taking the first derivative of the smoothed CD signal (using a 13-point Savitzky-Golay quadratic function) and finding the temperature at the peak or trough through visual inspection (associated with ± 0.5˚C error). The second method assumes a two-state model for DNA unfolding with constant ΔH and can be used for fully reversible melting transitions, i.e. when melting and cooling curves are (nearly) superimposable [25]. Hysteresis was determined as the difference between $T_m$ from the melting and cooling curves. Since the hysteresis never exceeded 3.3 ˚C, all systems were considered (nearly) reversible and the reported thermodynamic data were obtained using the two-state model. The results represent the average of 2–3 trials. All data manipulations were performed in Origin 2019b.

## Native Polyacrylamide Gel Electrophoresis (PAGE)

PAGE samples contained 40–50 μM DNA in 5K buffer and were weighted down with 7% w/v sucrose prior to loading. Twenty percent native polyacrylamide gels were made with 5 mM KCl and 1×Tris-Borate-EDTA. Gels were pre-migrated at 150 V for 30 min, loaded with 6–10 μL sample, and run for 120–150 min at 150 V at room temperature. A tracking dye was used to monitor gel progress and an oligothymidylate ladder consisting of dT$_{15}$, dT$_{24}$, dT$_{30}$, and dT$_{57}$ was used as a length marker. DNA bands were visualized using Stains-All and the resulting gel was captured with a conventional scanner.

## Analytical Ultracentrifugation (AUC)

AUC was carried out to determine the molecularity and purity of samples, as well as NMM binding stoichiometry. Samples of T1 and T7 were prepared at 1.3–5.0 μM either alone or with 1 or 5 eq. of NMM in 5K buffer. Samples of NMM alone at concentrations matching those in the DNA samples were used as controls. Sedimentation velocity measurements were carried out in a Beckman Coulter ProteomeLab XL-A analytical ultracentrifuge (Beckman Coulter Inc., Brea, CA) at 20.0˚C and 40,000 rpm in standard 2 sector cells. Two hundred scans were collected over a 10-hour centrifugation period at either 260 nm or 380 nm. Data were analyzed using Sedfit (www.analyticalultracentrifugation.com) [26] in the continuous c(*s*) mode. The sedimentation coefficient is denoted as *s*. Buffer density was determined on a Mettler/Paar Calculating Density Meter DMA 55A at 20.0˚C and buffer viscosity was measured on an Anton Paar Automated Microviscometer AMVn. For the calculation of molecular weight, 0.55 mL/g

was used for the partial specific volume. Binding of NMM to T1 and T7 was assessed by monitoring the appearance of an absorbance peak at 380 nm during the sedimentation run. Stoichiometry was determined as described earlier [27].

## Crystallography

Crystallization was achieved at room temperature using the hanging-drop vapor diffusion method. The T1-NMM sample was prepared by annealing HPLC-purified DNA with 1 eq. of NMM at 0.65 mM in 20K buffer. Drops were set manually at 2 μL DNA sample and 1 μL crystallization condition. The original crystals grew in condition 1–31 from the HELIX screen (Molecular Dimensions): 1.0 M sodium formate, 20% PEG 20000, and 0.05 M Bis-Tris pH 7.0. This condition was then optimized to 0.85 M sodium formate, 17.5% PEG 20000, and 0.05 M Bis-Tris pH 7.0. Large hexagonal crystals grew within 3 weeks to 300 μm in the largest dimension. Crystals were harvested and flash frozen in liquid nitrogen without additional cryoprotection.

The T7-NMM sample was prepared by annealing DNA with 1 eq. of NMM at 0.65 mM in 20K buffer. Drops were set by the TTP Labtech Mosquito Crystal liquid handler equipped with a humidity chamber at 0.1 μL DNA sample and 0.1 μL of the crystallization condition. Small hexagonal crystals grew within three weeks to 80 μm in the largest dimension from condition C5 of the Natrix screen (Hampton Research): 4.0 M LiCl, 0.01 M $MgCl_2$, and 0.05 M HEPES sodium pH 7.0. Crystals were cryoprotected in the base condition supplemented with 15% ethylene glycol before being flash frozen in liquid nitrogen.

Datasets were collected at the Advanced Photon Source 24 ID-E synchrotron facility to a maximum resolution of 2.39 Å for T1-NMM and 2.34 Å for T7-NMM. Raw diffraction data was processed using XDS [28]. The structures were solved by molecular replacement (MR) using PHENIX [29]. Three types of 3-quartet parallel GQ models were tested in MR: entire GQs; GQs with only thymine(s) in the loops (with other loop nucleotides removed); and GQ cores consisting of only G-quartets, with loops and overhangs removed. All models contained three $K^+$ ions. In addition, we also tried MR with a single G-quartet. The T7-NMM structure was solved using the GQ core from the structure of human telomeric DNA in complex with NMM (PDB ID: 4FXM) [30]. The initial MR solution was improved using PHENIX AutoBuild, and NMM was placed into the structure with PHENIX Ligand-Fit. Extensive manual model building cycles were performed in Coot [31] followed by PHENIX Refine. The T1-NMM structure was solved using T7 as the MR model, followed by AutoBuild, LigandFit, and manual model building, along with continuous refinement cycles. Data collection and refinement statistics are presented in Table 2.

The asymmetric units of both the T1-NMM and T7-NMM crystals contain two DNA chains (A and B), each of which is bound to one NMM molecule. The two DNA chains form a dimer in the case of T7 (i.e. A-B dimers), whereas they form dimers with symmetry-related molecules in the case of T1 (i.e. A-A' and B-B' dimers). Both structures include two $K^+$ ions within each GQ monomer, as well as a $K^+$ ion at the dimer interface. In the T1-NMM structure, the latter $K^+$ is at a special position, so it was modeled at 0.5 occupancy for each monomer. Due to a lack of clear electron density, the propionate groups of NMM were not built.

In the T1-NMM structure, the base for $T_5$ and $T_{14}$ in both chains and for $T_4$ and $T_{15}$ in chain B was not built due to loop disorder. Similarly, the sugar for $T_5$ in chain B was not built. NMM in chain B was modeled in two different orientations each at 0.5 occupancy with the *N*-Me group residing on different pyrrole nitrogens. A sodium formate molecule, present in the crystallization condition, was built near $G_{18}$ of chain B.

**Table 2. Crystallographic statistics for the T1-NMM and T7-NMM complexes.**

| | T1-NMM | T7-NMM |
|---|---|---|
| Resolution range (Å) | 64.78–2.388 | 51.34–2.339 |
| (Highest resolution shell) | (2.473–2.388) | (2.423–2.339) |
| Space group | R 3 2 | P 6₃ |
| Unit cell dimensions | | |
| a, b, c (Å) | 60.93, 60.93, 194.342 | 59.28, 59.28, 63.33 |
| α, β, γ (°) | 90, 90, 120 | 90, 90, 120 |
| Unique reflections | 5748 (492) | 5377 (498) |
| Redundancy | 19.0 (19.0) | 9.5 (6.1) |
| Completeness (%) | 97.48 (86.90) | 98.84 (92.39) |
| I/σ | 21.4 | 14.9 |
| R-merge (%) | 6.7 | 7.8 |
| R$_{work}$ / R$_{free}$ (%) | 23.52 / 24.91 | 20.01 / 22.43 |
| Number of atoms | 822* | 862 |
| DNA | 707 | 783 |
| NMM | 70 | 70 |
| Water | 0 | 4 |
| Potassium | 5 | 5 |
| Sodium formate | 4 | 0 |
| Copies in asymmetric unit | 2 | 2 |
| Overall B-factor (Å²) | 115.44 | 97.15 |
| RMS deviations | | |
| Bond lengths (Å) | 0.008 | 0.01 |
| Bond angles (°) | 0.83 | 1.2 |
| PDB ID | **6PNK** | **6P45** |

*NMM bound to chain B was modeled in two different orientations each at 0.5 occupancy. A potassium ion rests at a special position at the dimer interface and was modeled at 0.5 occupancy in each monomer. The total number of atoms, 822, in T1-NMM results from counting every atom of NMM in each of the two positions (105 atoms total), as well as an additional potassium (six total due to split occupancy of one K⁺).

In the T7-NMM structure, the base for T₁₅ in chain A and T₁₆ in chain B was not built due to loop disorder. In addition, disorder at the 5' and 3' overhangs of both chains resulted in poor electron density for the phosphate and base for T₁ and T₂₀, which were not built. Structure figures were prepared in PyMOL [32] and Coot. Atomic coordinates and structure factors have been deposited in the PDB under accession numbers **6PNK** (T1-NMM) and **6P45** (T7-NMM).

**Analysis of crystallographic data: G-quartet planarity, helical twist, torsional angles, RMSD, distances, groove widths, and B-factors.** G-quartet planarity, helical twist, and DNA backbone torsional angles were calculated following the methods described in our previous work [33]. Root mean square deviation (RMSD) was calculated by aligning all the atoms in each pairing of DNA chains from the T1-NMM and T7-NMM structures in PyMOL (with no outlier rejection). Distances between adjacent G-quartets and between the outermost G-quartet and NMM were calculated using the centroid of each G-quartet and of NMM (using the atomic coordinates of the 24 atoms comprising the porphyrin ring) with an in-house MATLAB script. Groove widths were measured in PyMOL as P-P distances, while B-factors were calculated using the Average_b PyMOL script (PyMOL Wiki, https://pymolwiki.org/index.php/Average_b).

## Results and discussion

### Biophysical characterization of constructs

Prior to crystallization, we characterized T2-T9 via TDS and CD to assess their similarity to T1. All constructs have TDS spectra with peaks at 240 and 275 nm and the characteristic trough at 296 nm (Fig 2A), indicating that they all form GQ structures. The GQ topology (e.g. parallel, mixed-hybrid, antiparallel [34]) was determined via CD scans (Fig 2B). A peak at 262 nm and trough at 241 nm is observed in all CD spectra, suggesting a parallel GQ topology. The similarity of the observed TDS and CD signatures for T1-T9 indicates that all constructs fold into similar secondary structures.

We then investigated the purity, homogeneity, and oligomerization state of all the constructs via PAGE (Fig 2C), as well as via AUC for T1 and T7 (Fig 3). PAGE reveals that among all constructs, T1 and T2 predominantly form dimers, although the streakiness of their bands likely suggests the presence of multiple species. AUC sedimentation velocity experiment on T1 confirms the presence of a dimer. Specifically, AUC shows two species, a monomer at 1.2 $S_{20,w}$ (6,880 Da) and 76 ± 12% dimer at 2.5 $S_{20,w}$, Fig 3A. Dimer formation by T1 was also detected earlier via size-exclusion HPLC, albeit to a lesser extent (11%), and by ESI-MS [10]. The observed discrepancy in the amount of dimer formed by T1 could be due to the difference in experimental conditions: we used a lower concentration of $K^+$ (5 vs. 100 mM) and DNA (2–5 vs. 250 μM) as well as different buffer (10 mM lithium cacodylate vs. 50 mM Tris-HCl). The rest of the constructs form monomers, with T3, T4, and T9 also having some amount of dimer. AUC of T7 (Fig 3A) supports the presence of at least 70% monomer (1.6 $S_{20,w}$, 7,500 Da),

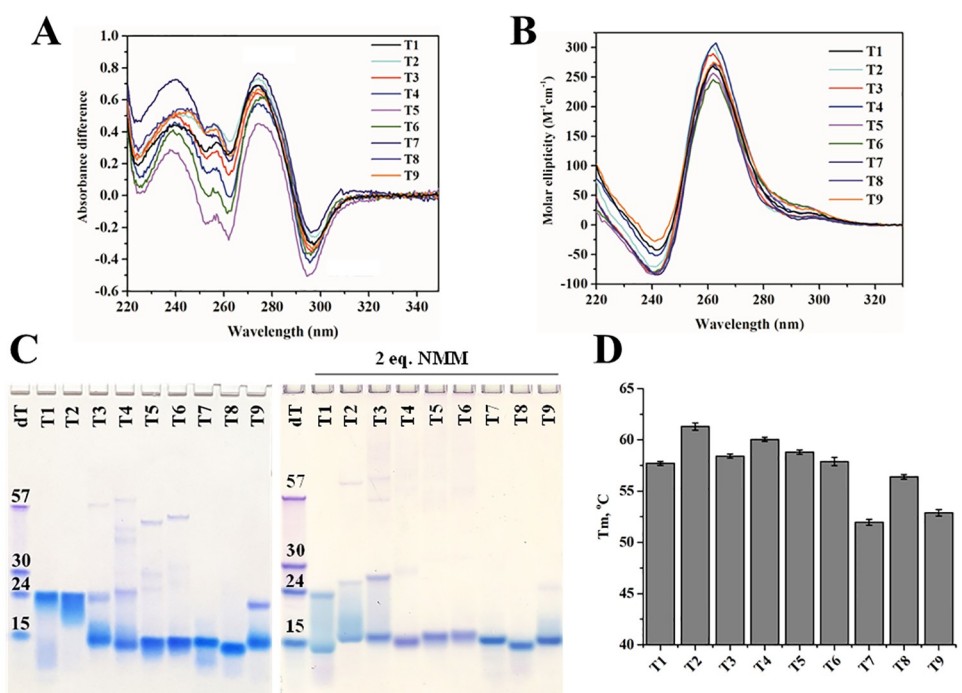

**Fig 2. Biophysical characterization of the T1–T9 constructs.** (**A**) TDS and (**B**) CD scans collected at 25 ˚C. (**C**) Twenty percent native PAGE gels for DNA alone (left) and annealed with 2 eq. of NMM (right). (**D**) $T_m$ for the constructs. All DNA samples were prepared in 5K buffer at ~5 μM except for the gel samples, which were prepared at 40–50 μM.

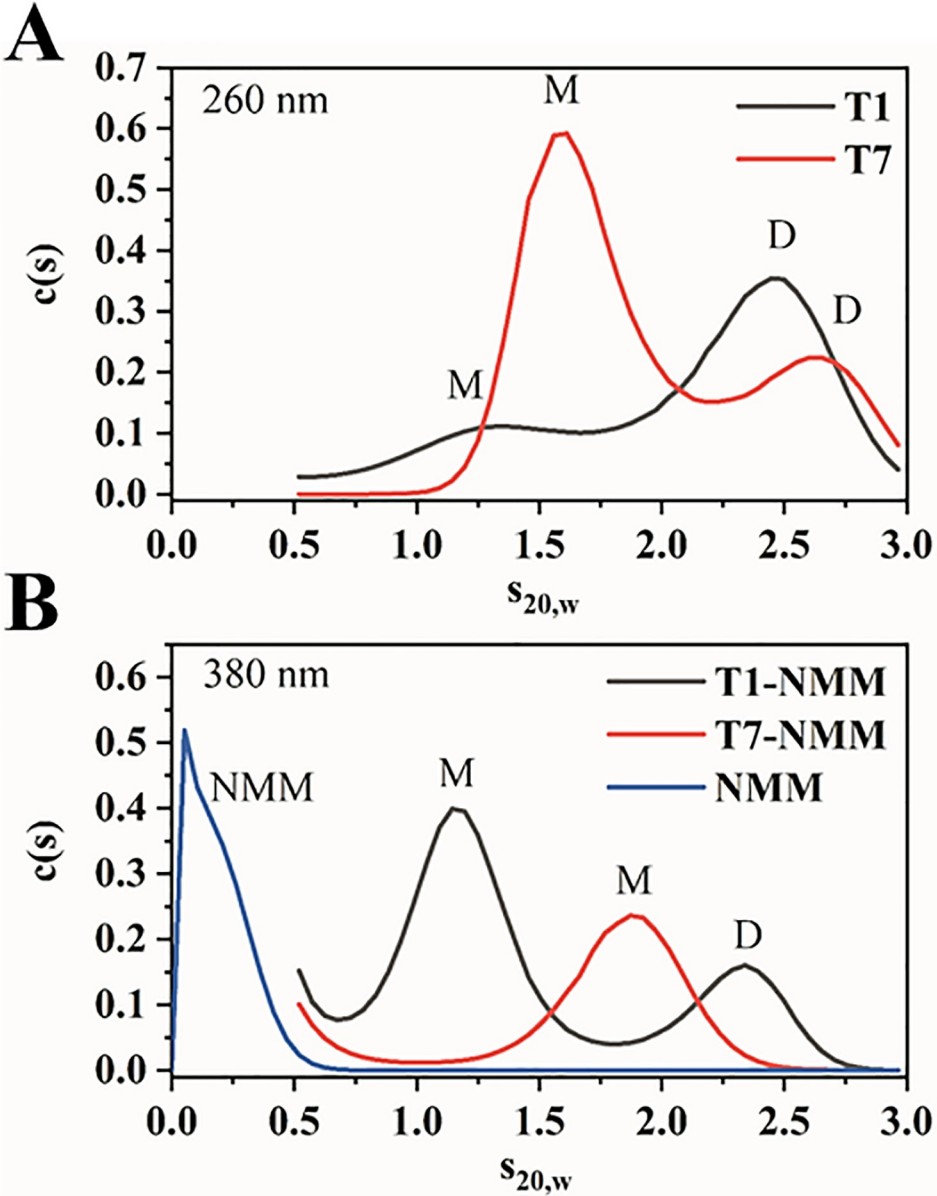

**Fig 3. AUC analysis of T1, T7, and their complexes with NMM.** AUC sedimentation velocity experiment provides estimates of the molecular weights of molecules along with their hydrodynamic shapes. (**A**) Representative results of sedimentation velocity measurements at 260 nm (monitoring DNA absorbance) for T1 and T7. At ~2 μM, T1 is a mixture of 27% monomer and 73% dimer, while T7 is 70% monomer and 30% dimer. (**B**) Representative results of sedimentation velocity measurements at 380 nm (monitoring NMM absorbance) in the absence and presence of T1 and T7. NMM alone sediments mainly as a small monomeric species with no aggregates in the size class of T1 and T7. In the presence of T1, NMM (5 eq.) sediments bound to monomeric (65%) and dimeric (35%) T1. In the presence of T7, NMM (5 eq.) sediments bound to a monomer only. Thus, NMM converts dimers of T1 and T7 to a monomeric state. Average stoichiometry was determined to be 1.0 ± 0.3 for T1-NMM (two trials) and 0.9 for T7-NMM (one trial). All samples were prepared in 5K buffer and the experiments were performed at 20˚C.

although some dimer (2.6 S$_{20,w}$) is evident. A previous ESI-MS study further confirms the monomeric state of T7 [10]. Finally, T3 –T6 also contain a small amount of higher order oligo-meric species (Fig 2C). Dimer formation in T1 –T4 and T9 can be explained by the presence of a 5' G, which often leads to dimerization [35–37].

It is curious that despite having a 5' G, the amount of dimer formed by T3 and T4 is small (Fig 2C). In these sequences, the 5' G-tract contains four or five Gs, while the rest of the G-tracts contain three Gs, thereby likely limiting the number of possible G-quartets to three. We hypothesize that the 5' Gs in excess of those required to form three G-quartets serve as flexible overhangs that break the dimer. However, dimer formation remains possible if one or two Gs from the 5' G-tract bulge out, engaging the 5' G in a G-quartet. In contrast, when a nucleotide other than guanine is found at the 5' end, no dimer formation is observed, as is the case for T5 –T8.

Next, we assessed the stability of all the constructs via CD melting studies (Fig 2D, Table 1). In all cases, the melting transitions were nearly reversible with low hysteresis ($< 3.3$ ˚C). T1 melts at $57.7 \pm 0.3$ ˚C in the presence of 5 mM $K^+$. Extending the T1 sequence has a mild effect on its stability, with the largest changes in $T_m$ being +3.6 for T2 and -5.8 ˚C for T7. Notably, all native constructs, T2 –T6, are slightly more stable than T1, likely due to stabilizing but non-essential capping interactions. Meanwhile, addition of non-native nucleotides in T7 –T9 destabilizes the structure, likely either due to increased flexibility (if the added nucleotides do not participate in stabilizing interactions) or disruption of the dimer. The stability of T1 and T7 under similar conditions was investigated by Largy *et al.* and yielded similar $T_m$ values [10].

It is interesting to compare T8 and T9, which have identical compositions but swapped C/G overhangs. T8 (with a 5' C) appears to be mostly monomeric with a $T_m$ of $56.4 \pm 0.3$ ˚C, while T9 (with a 5' G) is a mixture of monomer and dimer and displays a lower $T_m$ of $52.9 \pm 0.3$ ˚C. Thus, even the addition of seemingly unimportant overhang nucleotides affects both the oligomerization state and the stability of the resulting GQ, indicating that every nucleotide warrants consideration during construct design.

Apparent van't Hoff enthalpy values were estimated from the melting curves assuming a two-state equilibrium (Table 1). These values are at best estimates because only T7 and T8 form homogeneous monomolecular samples according to PAGE (Fig 2C). These enthalpy values dictate the slope of the melting curves near the $T_m$. Dimeric T1 and T2 display significantly higher ΔH values (~80 kcal/mol) as compared to T3-T8 (ΔH of ~50 kcal/mol) and T9 (ΔH of 38 kJ/mol). The higher enthalpies for dimeric GQs indicate that the dimers are maintained by a multitude of bonding interactions, which is consistent with our crystallographic results (presented below).

## Effect of NMM on T1-T9

NMM is a highly selective GQ binder with a clear preference for parallel GQ structures over other DNA folds [11]. Our laboratory demonstrated that NMM stabilizes T1 by an impressive $23 \pm 2$ ˚C at 2 eq. [9]. However, neither the binding strength nor the binding mode were previously established. To assess the effect of NMM on T1 –T9, we performed PAGE on DNA annealed in the presence of 2 eq. of NMM (Fig 2C). NMM increases the amount of monomeric GQ at the expense of dimers and higher order species. AUC data for T1 and T7 corroborate this finding, indicating that in the presence of 5 eq. of NMM, more than 65% of T1 adopts a monomeric state, while T7 is fully monomeric (Fig 3B). The slight upward shift in $S_{20,w}$ from 1.6 for T7 to 1.9 for T7-NMM indicates that NMM binding produces a hydrodynamically more compact complex with a reduced frictional coefficient, since the measured molecular weights of free and bound T7 are only slightly different.

According to PAGE, T7 and T8 form homogeneous complexes with NMM (Fig 2C) and are thereby good candidates for crystallographic studies. Thus, we characterized the T1-NMM, T7-NMM, and T8-NMM complexes via TDS, CD scans, and CD melting studies. The TDS signature of these complexes, while not similar to that of classical GQ DNA, still retains a small trough at 293 nm–indicative of a GQ fold–and displays a strong trough at 263 nm (S1A Fig).

This somewhat unusual TDS may result from interference due to NMM's absorbance. CD scans in the presence of NMM show that the signature of the parallel GQ is mostly unchanged. There is a small increase in the CD signal intensity for T7 and decrease for T1 and T8, and a small red shift of ~2 nm for all sequences (S1B–S1D Fig). Finally, CD melting studies demonstrate that NMM greatly stabilizes T1 by 20. ± 1˚C (consistent with our earlier work [9]), T7 by 19.9 ± 0.4 ˚C, and T8 by 17.0 ± 0.5˚C at 2 eq. (S2 Table). This stabilization is the highest reported in the literature for GQ-NMM complexes (see our recent review [11]). NMM had little effect on the value of ΔH (~55 kcal/mol).

## Spectroscopic characterization of NMM binding to T1 and T7

We used UV-vis and fluorescence spectroscopy to determine the binding constants and thermodynamic parameters for T1-NMM and T7-NMM. We performed Job plots and AUC to confirm binding stoichiometries.

**Binding studies via UV-vis and fluorescence titrations.** The UV-vis spectrum of NMM displays a 17.9 ± 0.4 nm red shift, low hypochromicity of 1 ± 4%, and an isosbestic point at 391 nm upon addition of T1 (Fig 4A). The values are nearly identical for T7-NMM with a red shift of 18.1 ± 0.5 nm, hypochromicity of -5 ± 4%, and an isosbestic point of 391 nm (S2A Fig). This data is consistent with values reported for UV-vis titrations of NMM with a variety of predominantly parallel GQs [9]. Titration data are best fit to the 1:1 binding model and yield a $K_a$ of 30 ± 20 μM$^{-1}$ for T1 (Fig 4B), signifying an impressively tight binding interaction. Binding of T7 to NMM is weaker, with a $K_a$ of 18 ± 7 μM$^{-1}$.

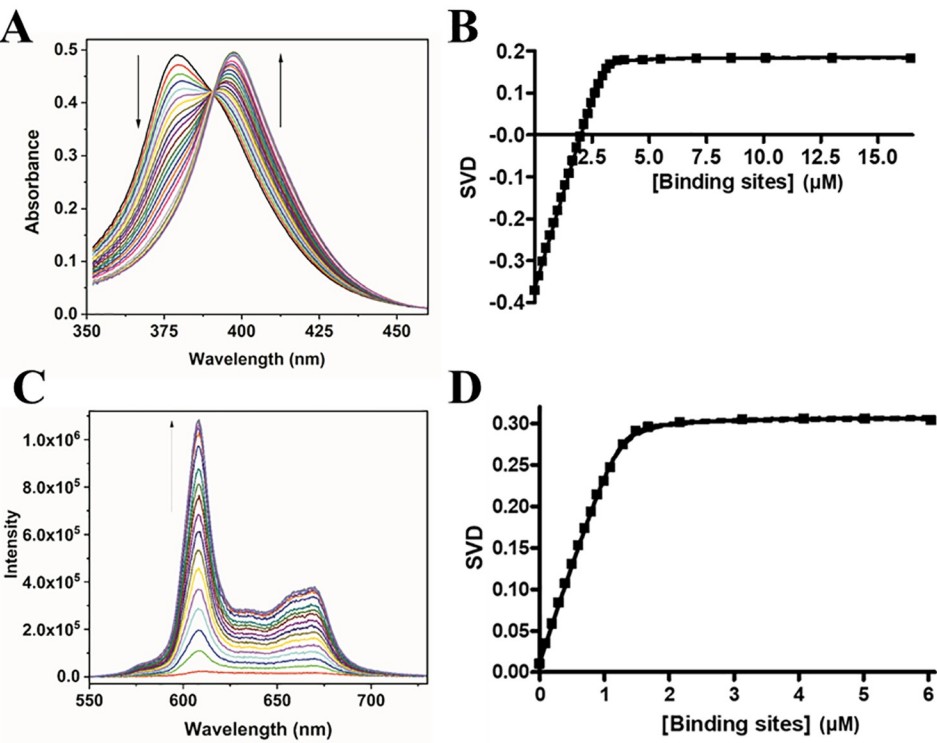

**Fig 4. Determination of $K_a$ for the T1-NMM complex via UV-vis and fluorescence titrations.** (**A**) Representative UV-vis titration of 3.4 μM NMM with 170 μM T1 to a final [T1]/[NMM] of 4.9 at 20 ˚C. (**B** and **D**) Fit of titration data (solid squares) to the 1:1 binding model with floating [NMM]. The 95% confidence interval is shown as dashed lines. (**C**) Representative FL titration of 1.0 μM NMM with 150 μM T1 to a final [T1]/[NMM] of 6.1 at 20 ˚C.

NMM displays a large increase in fluorescence, the so called 'light-switch' effect, in the presence of GQ DNA [12, 38]. We made use of this property to corroborate the $K_a$ for T1-NMM determined via UV-vis titrations. Fluorescence titration data yield a 1:1 binding stoichiometry with a $K_a$ of 50 ± 20 μM$^{-1}$ (Fig 4C and 4D). Fluorescence enhancement was found to be 49 ± 2, which is consistent with the fluorescence enhancement of 40–70 reported for a variety of parallel GQs (S3 Fig) [12]. The binding constants result in estimates of the binding free energy (ΔG = -RTlnK$_a$) of -10.3 and -9.9 kcal/mol for NMM binding to T1 and T7, respectively.

**T1 binds to NMM in a 1:1 stoichiometry determined via Job plot and AUC.** To confirm the T1-NMM binding stoichiometry, we used the model-independent Job plot method along with AUC. 1:1 binding in Job plot is signified by the mole fraction values close to 0.5. Our Job plot data shown in Fig 5 clearly and unambiguously suggest 1:1 binding, as does the result from AUC (Fig 3B), consistent with UV-vis and FL titration data.

In summary, our extensive binding studies demonstrate that NMM binds T1 with 1:1 stoichiometry and an extremely tight binding constant of 30–50 μM$^{-1}$. T7, with two additional terminal thymines, also binds NMM with 1:1 stoichiometry, but with a lower $K_a$ of 18 μM$^{-1}$.

## Characterization of the T1-NMM and T7-NMM complexes via X-ray crystallography

Biophysical characterization of T1 –T9 indicated that all the constructs form GQ structures with similar fold and stability to T1. We succeeded in producing diffraction quality crystals of

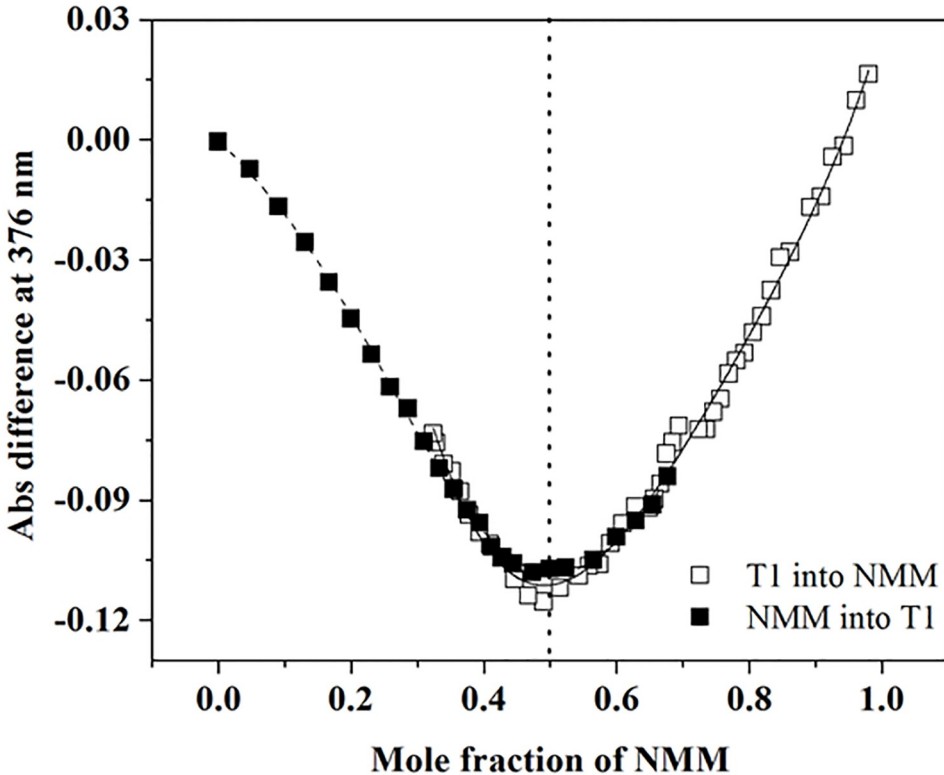

**Fig 5. Determination of stoichiometry for the T1-NMM complex via Job plot.** T1 and NMM were both prepared at 4.0 μM in 5K buffer at 20 ˚C, and the Job plot was built by plotting absorbance difference at 376 nm. Data at other wavelengths are consistent with the data presented here.

T1, T7, and T8 all with NMM, and solved the structure of the T1-NMM and T7-NMM complexes.

**Buffer selection for crystallization.** Buffer choice is critical for crystallization. The buffer should provide optimal stability for the biomolecule and should have simple composition with low ionic strength to avoid interfering with the crystallization process and overshadowing the components of the crystallization mixture. First, we investigated the effect of potassium on the fold and stability of T1. We conducted a CD melting study in 10 mM lithium cacodylate buffer pH 7.2 supplemented with 5–100 mM KCl. An increase in $K^+$ concentration led to a minor increase in CD signal intensity at 262 nm, higher thermal stability, and higher enthalpy of unfolding (S4 Fig). Melting of T1 was accompanied by a small hysteresis, which decreased from 3.3 ˚C at 5 mM KCl to 1.2 ˚C at 100 mM KCl. Next, we determined the effect of ionic strength on T1 stability by conducting the melting study in 5 mM $K^+$ buffer in the presence of 0 and 95 mM LiCl. T1 stability was not affected by the presence of LiCl ($T_m = 56.2 \pm 0.4$ and $57.7 \pm 0.3$ ˚C in 0 and 95 mM LiCl, respectively), but the enthalpy of T1 unfolding increased significantly from $47 \pm 1$ to $77 \pm 2$ kcal/mol. The data suggest that increased ionic strength facilitates stronger bonding within each monomer and between the monomers in the dimer, which otherwise repel each other due to the negatively charged DNA backbone.

Guided by the data, we chose 10 mM lithium cacodylate pH 7.2 and 20 mM KCl (20K) buffer for crystallographic studies. This buffer strikes the balance between assuring high thermal stability for T1 ($T_m$ of $65.5 \pm 0.9$ ˚C) and maintaining a low ionic strength of 30 mM.

**Atomic details of the T1-NMM and T7-NMM crystal structures.** Both T1-NMM and T7-NMM produced large hexagonal crystals (Fig 6A). The T1-NMM crystal structure was solved in the R 3 2 space group to 2.39 Å. The T7-NMM crystal structure was solved in the P $6_3$ space group to 2.34 Å and is characterized by an overall higher quality, so the subsequent discussion will be mainly focused on this structure. In both cases, the asymmetric unit contains two DNA chains, each of which binds one NMM molecule. In agreement with our biophysical data in solution (Fig 2B), the DNA folds into a parallel GQ in the crystalline state (Fig 6B) with all guanines adopting *anti* glycosidic bond conformations, as expected for a parallel GQ [39]. According to the GQ folding formalism [40], T1 and T7 can be classified as a type VIII GQs with four medium grooves. The average widths of the grooves are $16.1 \pm 0.1$ Å for T7 and $16.4 \pm 0.2$ Å for T1, S3 and S4 Tables. Interestingly, the 5' G-quartet–located at the dimer interface–has narrower grooves than the middle and 3' G-quartets, $15.5 \pm 0.2$, $16.2 \pm 0.2$, and $16.6 \pm 0.3$ Å, respectively, in T7-NMM.

The RMSD between the two chains is low, 0.919 Å for T1-NMM and 1.212 Å for T7-NMM (S5 Table), suggesting that the monomers within the asymmetric units are nearly identical. Furthermore, the T1 and T7 GQs are nearly identical to each other, with an average RMSD among each pairing of DNA chains of $1.3 \pm 0.4$ Å. Meanwhile, RMSD for the GQ core which consists of 12 guanines is $0.64 \pm 0.02$ Å, suggesting that the TT loops (particularly loops 1 and 3) are the main source of differences between the two GQ structures due to their flexibility.

T1-NMM and T7-NMM have high overall B-factors of 115.44 and 97.15 Å², respectively (S6 Table). The main contributors are flexible disordered loops, whose B-factors are 148.13 Å² for T1-NMM and 128.85 Å² for T7-NMM. T7 also contains 5' and 3' thymine overhangs with an average B-factor of 151.63 Å². The flexibility of these terminal thymines may explain the destabilization of T7 with respect to T1 by -5.8 ˚C (Table 1).

The G-quartets are spaced by 3.4 Å (S7 Table), leading to optimal π–π base stacking interactions. The average intramolecular helical twist between G-quartets within each GQ monomer is $29 \pm 4$˚ for T1 and $28 \pm 1$˚ for T7 (S8 Table), consistent with values reported for parallel GQs [39, 41] and calculated for the parallel Tel22-NMM structure ($29 \pm 1$˚) [30]. The two monomers are twisted with respect to each other by $116 \pm 1$˚, $117 \pm 2$˚ (S9 Table), and $120 \pm 3$˚, for

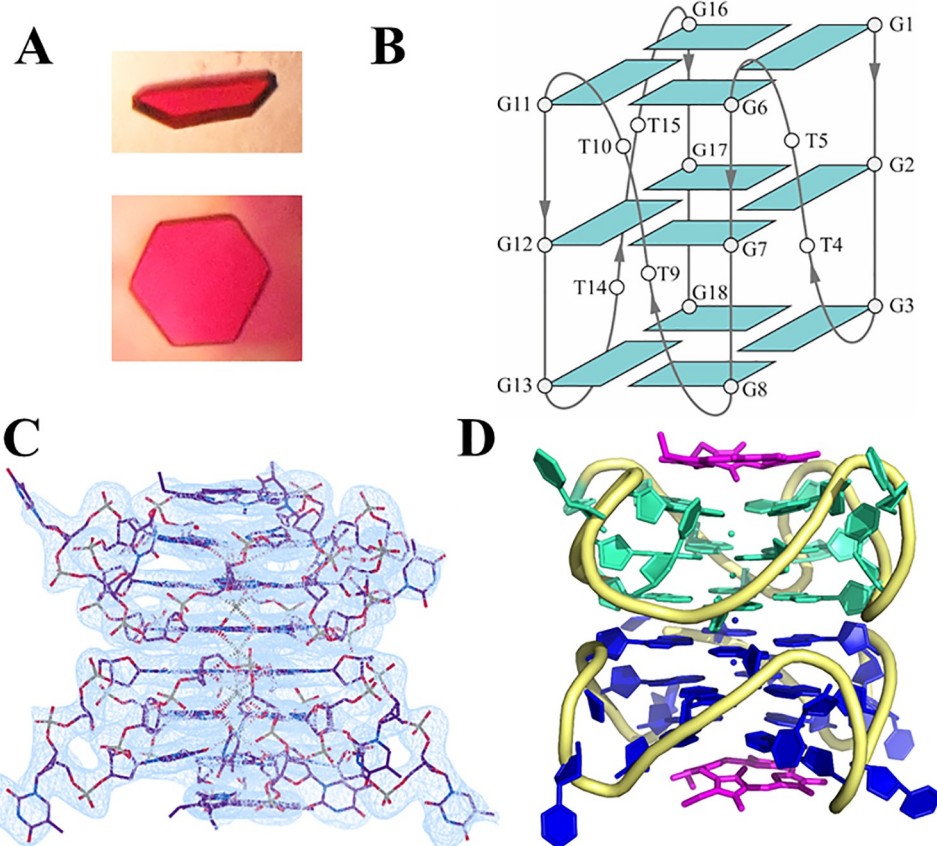

**Fig 6. Crystal structure of the T7-NMM complex.** (**A**) Representative crystal morphologies: half-hexagonal and hexagonal. (**B**) Schematic representation of the folding topology with the numbering scheme for T1. (**C**) Electron density at 1.0 I/σ surrounding the T7-NMM dimer. (**D**) The T7-NMM crystal structure reveals a 5'-5' GQ dimer capped at the 3' ends by NMM. Chain A is colored in teal, chain B is blue, the sugar-phosphate backbone is yellow, and NMM is pink. Potassium ions are depicted as spheres.

T1, T7 and Tel22, respectively. The DNA backbone torsional angles are plotted in S5 Fig and are consistent with those reported for other parallel GQ structures [39].

Both T1 and T7 GQs dimerize via their 5' G-quartets, and the dimers are capped on both 3' ends by NMM (Fig 6C and 6D). This arrangement is consistent with the 1:1 stoichiometry observed in UV-vis, FL, AUC, and Job plot experiments. A similar overall arrangement of GQs and NMM is observed in the crystal structure of human telomeric DNA, Tel22, with NMM (PDB ID: 4FXM) [30]. The GQ dimers are stabilized by five $K^+$ ions, all in square anti-prismatic coordination of the carbonyl oxygen from the eight guanines in the two adjacent G-quartets. Four $K^+$ ions are found between G-quartets within GQ monomers, while one $K^+$ is found at the dimer interface. The dimer in T7-NMM is further stabilized by base stacking between $T_5$ and $T'_{15}$ of chain B (where the prime (') notation signifies a symmetry-related molecule). There may be additional interactions between $T_5$ of chain A and either $T_{20}$ of chain B or $T'_{15}$ of chain A. However, the density for these two nucleotides is poor and neither nucleotide was fully built. Meanwhile, for T1-NMM, there may be stacking between $T_4$ in chain A and $T_{14}$ in chain B, but the electron density for $T_{14}$ is poor and its base was not built.

**Loop arrangement in the T1-NMM and T7-NMM structures.** The T1 and T7 GQs have three propeller TT loops. One thymine in each fully built loop is tucked into the groove, while

the other points out toward the solvent. A similar loop is seen in the NMR structure of d (TTGGGG)$_4$ in Na$^+$, which forms a three-quartet hybrid GQ with a propeller TT loop and lateral GTTG and TTG loops (PDB ID: 186D) [42]. As in our case, the propeller TT loop spans the medium groove and has one thymine tucked into the groove and another oriented into the solvent. The TT loop in the X-ray structure of a parallel GQ called TTLOOP (dGGGTGGGTGGGTTGGGTTAGCGTTA, loops underlined; PDB ID: 5DWW) has both Ts pointing into the solvent. One of those thymines forms intermolecular contacts with a single nucleotide T loop [43].

**Atomic details of the interaction between T1 or T7 GQ and NMM.** The central *N*-Me group of NMM in both T1- and T7-NMM is not well resolved, although there is some electron density for it. The propionic acid side chains of NMM are not visible and were not modeled; they were not visible in the Tel22-NMM structure (PDB ID: 4FXM), which was solved at a much higher resolution of 1.65 Å [30]. The lack of electron density even in the high-quality crystal structure signifies high flexibility of the propionic acid side chains and low likelihood of their engagement in the hydrogen bonding or ionic interaction with the GQ. The macrocycle of NMM is clearly defined.

NMM binds to the 3' G-quartet at a distance of 3.6 Å in T1-NMM, T7-NMM (S7 Table), and Tel22-NMM [30] structures. The distance was measured between the centroids of NMM and the 3' G-quartet. This distance is slightly longer than the optimal π-π stacking distance of 3.4 Å. The longer calculated distance may be partially due to the asymmetric distortion of NMM (*N*-Me bearing pyrrole ring carries most of the distortion in this molecule), which may skew the centroid used to calculate the distance. Nearly planar GQ ligands are typically located 3.2–3.6 Å from the terminal G-quartet. Examples include (PDB IDs in parentheses) berberine (3R6R [44]), acridines (3NZ7 [45], 1L1H [46], and 3EM2 [47]), daunomycin (3TVB [48]), salphen metal complexes (3QSC and 3QSF [49]), a ruthenium polypyridyl complex (5LS8 [50]), a dicarbene gold complex (5CCW [51]), and naphthalene diimides (3SC8 and 3T5E [52]).

The G-quartets become increasingly non-planar when moving away from the 5'-5' dimer interface. In this order, the out-of-plane deviations for the G-quartets in T7 are 0.39 ± 0.06, 0.97 ± 0.02, and 2.06 ± 0.06 Å. These numbers are similar to those determined for the Tel22-NMM structure (Table 3). Considering the significant non-planarity of the 3' terminal G-tetrad, we conclude that planar ligands may not be the best binding partners for parallel GQs. Indeed, we showed that the planar analogue of NMM, mesoporphyrin IX, does not bind Tel22 [9]. Instead, the non-planar terminal G-quartets in parallel GQs may bind optimally to somewhat distorted ligands like NMM. While the electron density for NMM in the T1-NMM and T7-NMM structures is not resolved enough to reliably calculate its overall non-planar deformation, NMM is known to be non-planar in its structure with Tel22 [30] and with the *Bacillus subtilis* wild type [53] and mutant [54] ferrochelatases.

In addition to stacking onto the 3' G-quartet, NMM interacts with T$_{10}$ from an adjacent DNA chain (Fig 7A and 7B for T7-NMM and S6 Fig for T1-NMM). This interaction leads to an interesting intermolecular assembly (Fig 7C and S6B Fig) that can explain the resulting space group (P 6$_3$ for T7-NMM and R 3 2 for T1-NMM) and the hexagonal shape of crystals (Fig 6A). For comparison, in the Tel22-NMM structure, NMM interacts with a 3' G-quartet on one face and an A'-T Watson-Crick base pair on the other face [30]. Thus, it is likely that the NMM-T interaction observed in T1-NMM and T7-NMM structures is a result of crystal packing forces.

**Crystal structures are good representations of the GQ-NMM complexes in solution.** To verify that the GQ conformation observed in the crystal structures is not an artifact of the crystallization process and that the high sample concentrations used did not affect GQ-NMM interactions, we compared the CD signatures and PAGE behavior of samples prepared for

**Table 3. Out-of-plane deviations (Å) for G-quartets in the T1-NMM, T7-NMM, and Tel22-NMM crystal structures.**

| G-Quartet | T1-NMM | | | T7-NMM | | | Tel22-NMM [30] |
|---|---|---|---|---|---|---|---|
| | Chain A | Chain B | Average | Chain A | Chain B | Average | Average |
| 5' | 0.42 | 0.37 | **0.40 ± 0.04** | 0.43 | 0.35 | **0.39 ± 0.06** | **0.49** |
| Middle | 0.57 | 0.36 | **0.5 ± 0.2** | 0.96 | 0.98 | **0.97 ± 0.02** | **1.08** |
| 3' | 1.65 | 1.78 | **1.72 ± 0.09** | 2.01 | 2.10 | **2.06 ± 0.06** | **1.85** |

crystallographic (~0.65 mM) and biophysical (~5 μM) studies. The similarity of the CD signatures (S7A Fig) indicates that the crystal structures are good representations of the GQ-NMM complexes found in solution. According to PAGE (S7B Fig), T1-NMM exists as a mixture of monomer and dimer regardless of concentration. Meanwhile, T7 is a monomer alone as well as in complex with NMM at both high and low concentrations. This data suggests that the T1 dimer observed in the crystal structure could exist under physiological conditions. However, the T7 crystallographic dimer is likely an artifact of crystal packing, wherein the thymine overhangs–normally freely moving in solution and potentially inhibiting dimerization–are tucked into the grooves, thereby allowing the 5' G-quartet to dimerize.

## Conclusion

We conducted detailed biophysical and crystallographic studies of the interaction between two G-rich DNA sequences, (GGGTT)₃GGG (T1) and (TGGGT)₄ (T7), and a highly selective porphyrin ligand, NMM. The T1 and T7 sequences are found in the human genome (including a gene implicated in Alzheimer's disease [14]); in the *T. thermophila* telomere (but do not represent the telomeric repeat); and in the genomes of at least 34 bacteria (including pathogenic species) [16, 17]. The DNA sequences fold into parallel GQs in solution both alone and in complex with NMM, as well as when bound to NMM in the crystal structures. The observed topology is consistent with a previous study that examined the fold of G-rich DNA with the

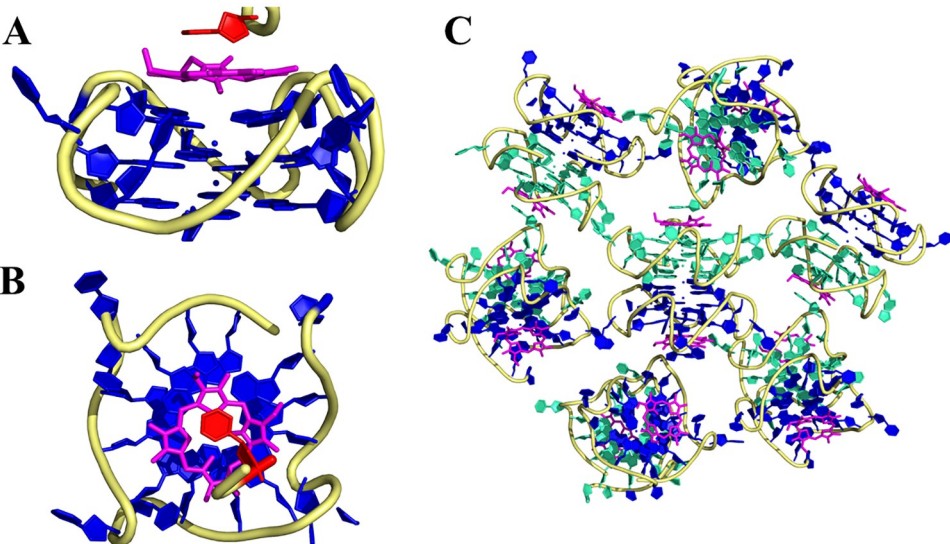

**Fig 7. Intermolecular interactions in T7-NMM. (A)** NMM (magenta) binds a GQ monomer (blue) via π-π stacking. NMM interacts with T₁₀ from another GQ monomer (red) on the other side. **(B)** Same as in **(A)** but rotated by 90 ˚. **(C)** Intermolecular interactions among T7 and NMM molecules. T7 monomers are depicted in teal and blue.

general sequence GGGT$_n$GGGT$_n$GGGT$_n$GGG, where n = 1–5 [55]. Short loops with 1–2 Ts, as is our case, lead to the parallel GQ conformation, whereas loops with 3 Ts lead to hybrid GQs and loops with 4–5 Ts lead to antiparallel GQs.

In solution, T1 exists mostly as a dimer of parallel GQs. Addition of thymines at the 5' and 3' ends or addition of NMM breaks the dimer but maintains the parallel GQ fold. The native extended constructs, T2 –T6, display similar stability and CD and TDS signatures to those of T1, indicating that our findings for the T1-NMM complex are applicable to longer DNA sequences with greater biological significance. Meanwhile, extending T1 with non-native nucleotides, as in T7 –T9, leads to some decrease in stability but preserves the overall GQ fold. Thus, using our DNA set as a representative example, we can conclude that GQ fold is determined by the number of nucleotides in G-tracts, as well as the number and nature of nucleotides in the loops. Meanwhile, the nature and length of 5' and 3' overhangs do not significantly affect GQ structures but can fine-tune their stability. Our earlier biophysical exploration of loop mutants designed based on the crystal structure of the antiparallel 19wt GQ from *Dictyostelium discoideum* [33] led us to conclude that loop interactions do not define GQ fold but rather fine-tune GQ stability, similarly to the effect of overhangs investigated here.

The binding of NMM to T1 is characterized by an unprecedentedly high binding affinity of 30–50 μM$^{-1}$, the strongest reported for any GQ-NMM complex. All reported binding affinities for GQ-NMM were discussed in our recent review [11]. We can explain the weaker binding of NMM to T7 ($K_a$ of 18 μM$^{-1}$) by the presence of terminal thymines that obstruct the binding surface at the 3' G-quartet. The impressive binding of NMM to T1 is also accompanied by 20 ˚C of stabilization at 2 eq. of NMM, the highest observed for any GQ [11]. NMM stabilizes T7 to the same extent. Thus, the binding affinity does not correlate with the stabilizing ability of a ligand, despite contrary expectation [56].

The impressively strong interaction between NMM and T1 is especially intriguing given that NMM does not display the characteristic features of a typical GQ ligand–planarity and a cationic nature. Instead, NMM is non-planar, with an observed out-of-plane deviation of 1.07 Å when free in solution [30] and is likely negatively charged under physiological pH due to its propionate groups. The non-planarity of NMM may be a key factor for its excellent binding affinity and selectivity for parallel GQs [11]. Our laboratory previously reported that the degree of NMM distortion varies based on its binding partner [30]. We also showed that the fully planar NMM derivative, mesoporphyrin IX, which lacks the central *N*-Me group, does not bind the human telomeric DNA GQ [9]. However, this observation may alternatively be due to the absence of the *N*-Me group rather than the planarity of mesoporphyrin IX.

In summary, our work identifies an unprecedentedly tight binding interaction between T1 and NMM, along with the highest reported stabilizing effect of NMM on a GQ. The crystal structures of T1-NMM and T7-NMM complexes reveal the end-stacking binding mode of the ligand. Our previous and current work demonstrates that the non-planar and negatively charged NMM is an excellent GQ binder. Although the GQ field is in search of planar, cationic GQ ligands, it is possible that ligands with some degree of non-planarity–or even better, the ability to alter their shape for an induced fit to their GQ binding partner–may allow for both tight binding to specific GQs and selectivity against dsDNA. Our work expands the small, but growing library of GQ-ligand crystal structures and provides atomic level information about GQ and ligand structural features that promote strong binding and stabilization. Our work also demonstrates how such binding can be detected and characterized thoroughly via biophysical and structural methods. The atomic coordinates reported here can be used to computationally search for even better drug candidates using available ligand libraries, with the hope that GQ binders may one day serve as anticancer therapeutics.

## Supporting information

**S1 Table. Occurrences of the T1 sequence in the human genome (GRCh38.p12 primary assembly) identified via BLAST.**
(DOCX)

**S2 Table. Thermodynamic stability of T1, T7, and T8 in the presence of 2 eq. of NMM in 5K buffer.**
(DOCX)

**S3 Table. Groove widths in the T7-NMM structure (Å).**
(DOCX)

**S4 Table. Groove widths in the T1-NMM structure (Å).**
(DOCX)

**S5 Table. RMSD (Å) for the T1- and T7-NMM structures.**
(DOCX)

**S6 Table. B-factors (Å$^2$) for the T1- and T7-NMM structures.**
(DOCX)

**S7 Table. Distances (Å) between G-quartets, between the outermost G-quartets and NMM, and between GQ monomers in the T1- and T7-NMM structures.**
(DOCX)

**S8 Table. Intramolecular helical twist (˚) between each quartet pair in the T1- and T7-NMM structures.**
(DOCX)

**S9 Table. Intermolecular helical twist (˚) at the dimer interface for T1-NMM and T7-NMM.** A′ signifies a symmetry related chain A molecule.
(DOCX)

**S1 Fig. CD and TDS signature of T1-NMM, T7-NMM, and T8-NMM.** (**A**) TDS and (**B-D**) CD scans for DNA alone at ~4 μM and with 2 eq. of NMM in 5K buffer at 20 ˚C.
(DOCX)

**S2 Fig. Determination of *K$_a$* for the T7-NMM complex via UV-vis titration.** (**A**) Representative UV-vis titration of 4.3 μM NMM with 163.5 μM T7 to final [T7]/[NMM] of 2.1. (**B**) Tit of titration data (solid squares) to the 1:1 binding model with floating concentration of NMM. The 95% confidence interval is shown as dashed lines.
(DOCX)

**S3 Fig. Fluorescence enhancement data for NMM in the presence of 10 eq. of the indicated DNA sequences.** T1 data (yellow) was collected with 2–5 eq. of DNA. Fig adapted from (12).
(DOCX)

**S4 Fig. Effect of K$^+$ concentration on fold and stability of T1.** Experiments were conducted with ~4.5 μM T1 in 10 mM lithium cacodylate pH 7.2 in the presence of 5–100 mM KCl. (**A**) CD signatures, (**B**) CD melting curves, (**C**) *T*$_m$, (**D**) ΔH, and (**E**) thermodynamic parameters determined from melting curves.
(DOCX)

**S5 Fig. Torsional angle wheel for the (A) T1-NMM and (B) T7-NMM structures.** The α, β, γ, δ, ε, and ξ angles characterize the DNA backbone while the χ angle indicates *syn* vs. *anti*

nucleotide conformation. Each individual angle is shown as a dot.
(DOCX)

**S6 Fig. Intermolecular interactions in the T1-NMM crystal structure.** (**A**) The T1-NMM asymmetric unit. (**B**) Intermolecular interactions among T1 and NMM molecules. As observed in the T7-NMM structure, NMM stacks onto the 3' G-quartet of one GQ and interacts with a thymine from another GQ (dashed box). Chain A of T1 is colored in teal, chain B is blue, the sugar-phosphate backbone is yellow, and NMM is magenta. Potassium ions are depicted as spheres.
(DOCX)

**S7 Fig. Comparison of crystallization and biophysical samples of T1- and T7-NMM.** Crystallization samples (cryst) contained 0.65 mM DNA and 0.65 mM NMM (1:1 DNA:NMM) in 20K buffer. Biophysical samples (biophys) contained 5 μM DNA and 10 μM NMM (1: DNA: NMM) if applicable in 5K buffer. (**A-B**) CD scans at 25 ˚C in 0.11 mm cuvettes (for cryst) or 10 mm cuvettes (for biophys). (**C**) Fifteen percent PAGE. Cryst samples were diluted to 50 μM immediately before loading while biophys samples were prepared at 50 μM. The gel was run with 5 mM KCl at 150 V for 150 min at room temperature and visualized using Stains-All.
(DOCX)

**S1 Raw images.**
(PDF)

## Acknowledgments

We would like to thank Adam Olia (University of Pennsylvania), Daniela Fera (Swarthmore College), and Jack Nicoludis (University of California San Francisco) for their generous help and advice in our crystallographic studies. This work is based on research conducted at the Northeastern Collaborative Access Team using the Eiger 16M detector on 24-ID-E beam line. This research used resources of the Advanced Photon Source, a U.S. Department of Energy (DOE) Office of Science User Facility operated for the DOE Office of Science by Argonne National Laboratory.

## Author Contributions

**Conceptualization:** Liliya A. Yatsunyk.

**Data curation:** Linda Yingqi Lin, Sawyer McCarthy, Yanti Manurung, Irene M. Xiang, William L. Dean.

**Formal analysis:** Linda Yingqi Lin, Sawyer McCarthy, Barrett M. Powell, Yanti Manurung, Irene M. Xiang, William L. Dean, Brad Chaires, Liliya A. Yatsunyk.

**Funding acquisition:** Liliya A. Yatsunyk.

**Project administration:** Liliya A. Yatsunyk.

**Supervision:** Liliya A. Yatsunyk.

**Validation:** William L. Dean, Brad Chaires.

**Writing – original draft:** Linda Yingqi Lin, Liliya A. Yatsunyk.

**Writing – review & editing:** Linda Yingqi Lin, Sawyer McCarthy, William L. Dean, Brad Chaires, Liliya A. Yatsunyk.

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
