## [Decision Letter · Decision Letter 0]

11 Sep 2020

PONE-D-20-22649

Biophysical and X-ray structural studies of the (GGGTT)3GGG G-quadruplex in complex with N-methyl mesoporphyrin IX

PLOS ONE

Dear Dr. Yatsunyk,

Thank you for submitting your manuscript to PLOS ONE. After careful consideration, we feel that it has merit but does not fully meet PLOS ONE’s publication criteria as it currently stands. Therefore, we invite you to submit a revised version of the manuscript that addresses the points raised during the review process.

We apologize for the delay encountered in the review process. Your manuscript was determined to be sound and solid by two experts in the field. One reviewer noted minor issues that need to be addressed before the manuscript can be published, in particular the quality of figures 6 and 7.  

We look forward to receiving your revised manuscript.

Kind regards,

Michel M Ouellette, Ph.D.

Academic Editor

PLOS ONE

Journal Requirements:

3.PLOS ONE now requires that authors provide the original uncropped and unadjusted images underlying all blot or gel results reported in a submission’s figures or Supporting Information files. This policy and the journal’s other requirements for blot/gel reporting and figure preparation are described in detail at https://journals.plos.org/plosone/s/figures#loc-blot-and-gel-reporting-requirements and https://journals.plos.org/plosone/s/figures#loc-preparing-figures-from-image-files. When you submit your revised manuscript, please ensure that your figures adhere fully to these guidelines and provide the original underlying images for all blot or gel data reported in your submission. See the following link for instructions on providing the original image data: https://journals.plos.org/plosone/s/figures#loc-original-images-for-blots-and-gels.

4.Thank you for stating the following in the Acknowledgments Section of your manuscript:

[This work is supported by the National Institutes of Health

[grant number 1R15CA208676-01A1 to LY; GM077422 to BC and WD] and Henry Dreyfus

35

Teacher-Scholar award to LY. This work is ba 695 sed upon research conducted at the Northeastern

Collaborative Access Team beamlines, which are funded by the National Institute of General

Medical Sciences from the National Institutes of Health (P30 GM124165). The Eiger 16M

detector on 24-ID-E beam line is funded by a NIH-ORIP HEI grant (S10OD021527).]

 [The funders had no role in study design, data collection and analysis, decision to publish, or preparation of the manuscript.]

Reviewers' comments:

Reviewer's Responses to Questions

**Comments to the Author**

1. Is the manuscript technically sound, and do the data support the conclusions?

Reviewer #1: Yes

Reviewer #2: Yes

2. Has the statistical analysis been performed appropriately and rigorously? 

Reviewer #1: N/A

Reviewer #2: N/A

3. Have the authors made all data underlying the findings in their manuscript fully available?

Reviewer #1: Yes

Reviewer #2: Yes

4. Is the manuscript presented in an intelligible fashion and written in standard English?

Reviewer #1: Yes

Reviewer #2: Yes

5. Review Comments to the Author

Reviewer #1: The manuscript by Yatsunyk and coworkers deals with the study of the interaction of a G-quadruplex (G4) forming sequence GGGTTGGGTTGGGTTGGG (T1), with a water-soluble porphyrin, N-methyl mesoporphyrin IX (NMM) via biophysical and X-ray crystallographic studies. The authors have demonstrated that the interaction has a 1:1 binding stoichiometry with a tight binding constant. Eight variants of T1 were also characterized and the crystal structures of the T1- and T7-NMM complexes were determined. This manuscript provides details about G4-ligand binding interactions and informs the design of novel anticancer drugs, for these reasons the manuscript deserves publication in PLOS one

Reviewer #2: The study of Yatsunik et al. represents an exhaustive in solution and solid state study of GQ constructs, two of them in complex with NMM. The boiphysical chachterization is complete and the investigation accurate, the manuscript is fluent and well organized

Table 1: the authors should add the indications of the 5’ and 3’ ends

p.4-5: the authors should better stress the relevance of the studied constructs in human

p.5-7, p.36-37: it seems that the sequence of the reference numbering is incorrect. Please check

p.5 l.95: the search in the pdb database using the key words and refinements parameters reported, provided an higher numbers of entries

p.16 Table 2: the authors should explain why the ligand (NMM) has different atom numbers in the two structures

p.27 l.535-537: please rephrase

p.28 l.578-580: please specify the distance

The quality of Figures 6 and 7 is poor. Please provide pictures of higher quality

6. PLOS authors have the option to publish the peer review history of their article (what does this mean?). If published, this will include your full peer review and any attached files.

Reviewer #1: No

Reviewer #2: No

---

## [Author Response · Author response to Decision Letter 0]

5 Oct 2020

Response to Reviewers

We thank both Reviewers for their positive feedback and useful suggestions. 

Table 1: the authors should add the indications of the 5’ and 3’ ends

Added 5' and 3' to the header row of Table 1.

p.4-5: the authors should better stress the relevance of the studied constructs in human

We have restructured and expanded the relevant paragraph to stress the relevance of the T1 construct in human:

“We used BLAST (13) to align T1 with a number of genomes. The T1 sequence occurs 11 times in the human genome in seven distinct regions across six chromosomes including within the zinc finger protein 292 gene; on the complementary strand of the plexin A4 gene; within the C-type lectin domain family 2 member A gene; carboxypeptidase M gene, fas binding factor 1 gene; and LOC107984696 non-coding RNA (Table S1). T1 may also be involved in regulating β-secretase 2 (BACE2), a gene implicated in Alzheimer’s disease (17). In addition, the T1 sequence is found at position 168,273 in the Tetrahymena thermophila genome in the telomeric region. However, it does not represent the common telomeric motif, (GGGGTT)n (crystallographic studies on this consensus motif are underway in our laboratory). Recently, a protein that binds parallel telomeric GQs (Kd = 11.5 μM) was discovered in T. thermophila (16), thereby providing compelling evidence for the biological relevance of such GQ structures. Finally, T1 sequence is found in the bacteria Neisseria meningitidis, Neisseria gonorrhoeae, the bird pathogen Escherichia coli strain APEC O78, and repetitively in Paenibacillus (18,19).”

We have done the same in the Conclusion part:

“The T1 and T7 sequences are found in the human genome (including a gene implicated in Alzheimer’s disease (14)); in the T. thermophila telomere (but do not represent the telomeric repeat); and in the genomes of at least 34 bacteria (including pathogenic species) (16,17).”

p.5-7, p.36-37: it seems that the sequence of the reference numbering is incorrect. Please check

Checked and corrected

p.5 l.95: the search in the pdb database using the key words and refinements parameters reported, provided a higher numbers of entries

We performed the search with the key words “"G-quadruplex Ligand". Bulk of the entries contained K+, Na+, and Mg2+ as ligands, and those were removed. Out of the other entries, only unique structures were counted. For example, there are two structures of human telomeric DNA with N-methylmesoporphyrin IX from our lab but we counted them as one entry only. 

We have added the following clarifying sentence:

These numbers were obtained by searching PDB for "G-quadruplex Ligand", refining the search parameters by "DNA" and either "X-ray Diffraction" or "Solution NMR", and manually examining all entries keeping those that are unique.

p.16 Table 2: the authors should explain why the ligand (NMM) has different atom numbers in the two structures

All atoms in the T7-NMM structure were modeled at full occupancy, whereas one NMM molecule and one potassium ion in the T1-NMM structure were modeled in two different positions each at 0.5 occupancy. In Table 2 every atom in NMM was counted, including those modeled in two positions, leading to doubling the numbers. In addition, “ligand” in Table 2 refers to NMM, potassium ions and sodium formate molecules. Below is a detailed breakdown of the identities of the atoms counted in the Table 2:

 T1-NMM T7-NMM

NMM bound to chain A

(built without propionate groups) 35

 35

NMM bound to chain B

(built without propionate groups) 35+35

(NMM in chain B was modeled in two different orientations each at 0.5 occupancy) 35

Total NMM atoms 105 70

Potassium ions 6

(There are five K+ with the one at the dimer interface modeled at 0.5 occupancy) 5

Sodium formate 4 0

Total ligand atoms 115 75

Based on the comment from the reviewer, we have revised Table 2 to clearly indicate that both structures have two NMM molecules and five potassium ions: 

Table 2. Crystallographic statistics for the T1-NMM and T7-NMM complexes.

Number of atoms 822* 862

 DNA 707 783

 NMM 70 70

 water 0 4

 Potassium 5 5

 Sodium formate 4 0

*NMM molecule bound to chain B was modeled in two different orientation each at 0.5 occupancy; one potassium ion rests on the special position at the dimer interface and was modeled at 0.5 occupancy in each monomer. The total number of atoms, 822, in T1-NMM results from counting every atom of NMM in each of the two position as well as an additional potassium (five total, counted as six due to split occupancy of one K+).

p.27 l.535-537: please rephrase

(Preceding sentence: “In addition, the 5’ and 3’ thymine overhangs in T7 (not present in T1) are highly disordered, with a B-factor of 151.63 Å2.”)

“The flexibility of these terminal thymines, along with the potential of these overhangs to destabilize the dimer observed in solution for T1 but not T7, may explain the destabilization of T7 with respect to T1 by -5.8 °C (Table 1).”

We have rephrased the whole paragraph in the following way:

T1-NMM and T7-NMM have high overall B-factors, 115.44 and 97.15 Å2, respectively (Table S5). Main contributors are flexible disordered loops whose B-factors are 148.13 Å2 for T1-NMM and 128.85 Å2 for T7-NMM. T7 also contains 5’ and 3’ thymine overhangs with an average B-factor of 151.63 Å2. The flexibility of these terminal thymines may explain the destabilization of T7 with respect to T1 by -5.8 �C (Table 1). 

p.28 l.578-580: please specify the distance

“NMM binds to the 3’ G-quartet at a distance of 3.6 Å in both the T1- and T7-NMM structures (Table S6), as well as in the Tel22-NMM structure (28). This distance is slightly longer than the optimal π-π stacking distance of 3.4 Å.”

We have specified the distance in the experimental section:

Distances between adjacent G-quartets and between the outermost G-quartet and NMM were calculated using the centroid of each G-quartet and of NMM (using the atomic coordinates of the 24 atoms comprising the porphyrin ring).

We have also added the following sentence to the Result section for clarification:

This distance was measured between the centroids of NMM and the 3’ G-quartet.

The quality of Figures 6 and 7 is poor. Please provide pictures of higher quality

We have provided higher quality figures.

---

## [Decision Letter · Decision Letter 1]

16 Oct 2020

Biophysical and X-ray structural studies of the (GGGTT)3GGG G-quadruplex in complex with N-methyl mesoporphyrin IX

PONE-D-20-22649R1

Dear Dr. Yatsunyk,

We’re pleased to inform you that your manuscript has been judged scientifically suitable for publication and will be formally accepted for publication once it meets all outstanding technical requirements.

Kind regards,

Michel M Ouellette, Ph.D.

Academic Editor

PLOS ONE

Additional Editor Comments (optional):

Reviewers' comments:

Reviewer's Responses to Questions

**Comments to the Author**

1. If the authors have adequately addressed your comments raised in a previous round of review and you feel that this manuscript is now acceptable for publication, you may indicate that here to bypass the “Comments to the Author” section, enter your conflict of interest statement in the “Confidential to Editor” section, and submit your "Accept" recommendation.

Reviewer #1: All comments have been addressed

Reviewer #2: All comments have been addressed

2. Is the manuscript technically sound, and do the data support the conclusions?

Reviewer #1: Yes

Reviewer #2: Yes

3. Has the statistical analysis been performed appropriately and rigorously? 

Reviewer #1: N/A

Reviewer #2: N/A

4. Have the authors made all data underlying the findings in their manuscript fully available?

Reviewer #1: Yes

Reviewer #2: Yes

5. Is the manuscript presented in an intelligible fashion and written in standard English?

Reviewer #1: Yes

Reviewer #2: Yes

6. Review Comments to the Author

Reviewer #1: The authors have addressed all referees' concerns. The manuscript has been improved and it is ready for publication.

Reviewer #2: All comments have been addressed. I have no more comments. The manuscript is now acceptable for publication

7. PLOS authors have the option to publish the peer review history of their article (what does this mean?). If published, this will include your full peer review and any attached files.

Reviewer #1: No

Reviewer #2: No

---

## [Editor Report · Acceptance letter]

26 Oct 2020

PONE-D-20-22649R1 

Biophysical and X-ray structural studies of the (GGGTT)_3_GGG G-quadruplex in complex with *N*-methyl mesoporphyrin IX 

Dear Dr. Yatsunyk:

I'm pleased to inform you that your manuscript has been deemed suitable for publication in PLOS ONE. Congratulations! Your manuscript is now with our production department. 

Kind regards, 

on behalf of

Dr Michel M Ouellette 

Academic Editor

PLOS ONE